# Population Genetic Structure of Historic Olives (*Olea europaea* subsp. *europaea*) from Jordan

**DOI:** 10.3390/ijms262210863

**Published:** 2025-11-09

**Authors:** Nawal Alsakarneh, Aseel Abu Kayed, Fadwa Hammouh, Hamad A. Alkhatatbeh, Maysoun S. Qutob, Bayan Alkharabsheh, Wisam M. Obeidat, Ahmad Ateyyeh, Monther T. Sadder

**Affiliations:** 1Department of Nutrition and Food Processing, Al-Huson University College, Al-Balqa Applied University, P.O. Box 50, Huson, Irbid 21510, Jordan; aseel.abukayed@bau.edu.jo; 2Nutrition and Heath Psychology Department, Faculty of Health Sciences, American University of Madaba, Madaba 11821, Jordan; f.hammouh@aum.edu.jo; 3Ash-Shoubak University College, Al-Balqa Applied University, Al-Salt 19117, Jordan; 4Department of Clinical Nutrition and Dietetics, Faculty of Allied Medical Sciences, Applied Science Private University, Amman 11937, Jordan; m_alqutob@asu.edu.jo; 5Department of Horticulture and Crop Science, School of Agriculture, University of Jordan, Amman 11942, Jordan; radwan_bayyan22@yahoo.com (B.A.); ateyyeha@ju.edu.jo (A.A.); 6Department of Plant Protection, School of Agriculture, University of Jordan, Amman 11942, Jordan; wi.obeidat@ju.edu.jo

**Keywords:** *Olea europaea*, historic olive, ISSR primers, dendrogram, genome structure, Jordan

## Abstract

Major historic olive tree cultivars around the Mediterranean originate from the Jordan area and possess a proven abiotic stress tolerance; however, they were unexplored from the diversity perspective. Therefore, historic olive tree accessions from three northern regions—Irbid (i), Jerash (J), and Ajloun (A)—were analyzed using DNA molecular markers to identify and study their genetic relationships and genetic structure. DNA molecular markers of inter-simple sequence repeats (ISSR) were used. A total of 3150 data entries (859 present and 2291 absent) were generated with fragment sizes ranging from 350 to 2000 bp. Data entries were evaluated with UPGMA and population genetic structure analysis. The results showed that similarity among the investigated sixty-three accessions ranged from 9% between J14 and i20 up to 100% between ‘J11’ and ‘J12’ and between A8 and A9. The discriminating power values for ISSR_807, ISSR_810, and ISSR_825 were 0.70, 0.61, and 0.83, respectively. A generated dendrogram showed ten major clades, while the genetic structure could resolve four unique genetic pools: one for Irbid, one for Jerash, and two for Ajloun. In addition, analysis of 19 phenotypic parameters covering leaf, fruit, stone, and flesh was able to confirm the molecular data. Phenotypic and ISSR data were analyzed using PCA, cluster, and Mantel tests. ISSR markers showed clear genetic differentiation among groups, whereas phenotypic traits displayed lower variation but a significant correlation with molecular diversity. Promising accessions with either pure or admixture genetic makeup were identified. The resolved genetic structure of the investigated historic olive accessions would open new frontiers for olive breeding and utilization, helping to overcome current production challenges and climate change limitations.

## 1. Introduction

Olives are members of the Oleaceae family, which has about 600 species and 30 genera. Cultivated olives are found throughout the Mediterranean and are members of the *Olea* genus, which includes the *europaea* species and the *sativa* subspecies. *Olea europaea* is thought to have evolved because of its attractive natural traits [1]. Jordan’s olive ecosystem is considered a national heritage and among the world’s oldest. Jordan, a nation in the eastern Mediterranean basin, is among the world’s top olive cultivation countries, with over 11 million olive trees that produced roughly 35,828 tons of olive oil in the 2024/2025 season [2]. According to a study conducted by French and Jordanian researchers, the village of Hadeib Al-Reeh in Wadi Rum, Jordan, may be the oldest place on Earth where olive trees have been grown, with evidence of their presence dating back more than 7500 years [3]. Olives are thought to have been domesticated as early as 6200 BCE in Jordan, with signs of oil pressing in Pella, in the northern Jordan Valley, dating back to at least 5200 BCE [4], while olive cultivation first appeared during the Chalcolithic period (around 5400 BC), according to an analysis of ashes from fireplaces in the settlement remains [5]. Recent studies explored olive oil production during the Early Bronze Age (3800–2000 BCE) in the southern Levant, north of Jordan [6]. This implies that Jordan was a major hub for olive domestication in the Mediterranean.

One of the oldest genetic olive genotypes in the Mediterranean region is ‘Mehras,’ a historic olive cultivar from the Maysar region in the municipality of Alhashemya in the Ajloun province. ‘Mehras’ was chosen rather than Romi because the local cultural tradition, particularly in Ajloun, makes a distinction between the size of olive trees [7]. As a national geographical identifier and a brand linked to 1000-year-old living olives in Jordan, the historic ‘Mehras’ olive cultivar has received unprecedented attention [7]. ‘Mehras’ is a true ancestral variety that has endured over time. Its genetic fingerprint has demonstrated a rich and distinct genetic diversity among available olive varieties in the Mediterranean region [8]. Because of its long lifespan and the low rate of genotype turnover brought about by centuries of culture, the cultivated olive tree is an evergreen, outcrossing, vegetatively propagated tree with a vast genetic patrimony. In addition to the numerous instances of similar and synonymous names, the enormous number of cultivars makes it challenging to describe and categorize different types of olives [9]. ‘Mehras’’ capacity to adapt to severe situations and climate change while preserving its unique characteristics is greatly influenced by its genetic characteristics and the quality of olive oil [7].

There is debate over how many genotypes of olives are cultivated globally. According to the Olive Tree Germplasm Database, there are 1275 genotypes known to exist worldwide, which are grown in 54 countries and conserved in 100 live gene bank collections. However, due to the limited knowledge on certain cultivars and ecotypes, the number of olive varieties may be higher than previously reported [10]. Southern European nations like Italy (538 variations), Spain (183), France (88), and Greece (52) are home to the majority of these cultivars [11]. In addition to numerous international genotypes, local cultivars are grown across Jordan [12,13]; these are thought to be the best since they are high-quality, resilient to the region’s climate, and can tolerate drought [14] and salinity stresses [15].

In Jordan two-thirds of cultivated olives are of the historic type; they have been grown for hundreds of years in sporadic orchards in all villages across the country. They are mainly grown in uphill regions where they are mainly rain fed. Therefore, they are resilient to drought and diseases [7,14]. These trees form precious genetic pools with unprecedented potential for olive cultivation and improvement. However, major pan-Mediterranean studies of cultivated olive cultivars have unfortunately ignored these historic genotypes. In one of the iconic olive diversity studies, 289 olive cultivars from all counties around the Mediterranean basin were genotyped at 25 simple sequence repeat (SSR) loci. The sample represented genetic diversity at all available geographic extremes [16]. However, no single cultivar from Jordan including historic genotypes was included. Likewise, in a more recent pan-Mediterranean diversity study, a genomic variation map and the most comprehensive catalog/resource of molecular variation were achieved for 76 olive tree cultivars originating from the entire basin [17], and again excluded any cultivar from Jordan. This presents a major research gap to investigate these historic genotypes.

Scientists have used different methods to identify olive cultivars as morphological; recently DNA molecular markers have been used to identify olive cultivars. Olive cultivars’ fingerprinting and genetic relatedness patterns were usually determined by DNA molecular techniques [13]. Various molecular markers have been used in efforts to conserve olive genetic resources and utilized in breeding, including single nucleotide polymorphism (SNP) [18], simple sequence repeat (SSR) [19], inter-simple sequence repeat (ISSR) [13], amplified fragment length polymorphism (AFLP) [20], restriction fragment length polymorphism (RFLP) [21], and random amplified polymorphic DNA (RAPD) [12]. The technique, which is based on the amplification of DNA segments between two microsatellite repeated regions, was created in response to the requirement to investigate microsatellite repeats without the use of DNA sequencing [22]. ‘Mehras’ is a true ancestral variety that has endured over time; similarly, additional historic accessions along the Jordan may carry related features. In fact, the ‘Mehras’ genome has demonstrated a rich and distinct genetic diversity among the available olive varieties in the Mediterranean region [23].

ISSR is a straightforward, quick, and effective method that yields amplified products with lengths ranging from 200 to 2000 bp. Because longer primers enable higher annealing temperatures, the method is very reproducible [22]. Numerous noteworthy features of the ISSR methodology include its PCR-based nature, reproducibility, dominance, locus-specificity, high degree of polymorphism, and ability to be utilized for fingerprinting and germplasm conservation. Each PCR requires a single primer [13]. PCR amplification utilizing an ISSR primer and extracted genomic DNA as a template is typically the first step in ISSR analysis. Agarose or polyacrylamide gel electrophoresis of the PCR products and visualization come after ISSR-PCR. The ISSR bands are then scored, and data analysis comes last [13]. The primary benefit of ISSRs is that primer sequences can be determined without any prior knowledge of DNA data. Furthermore, the use of PCR enables the use of a very tiny amount of DNA in ISSR analytical processes. For each reaction, 10–50 ng of high-quality DNA is usually enough [22].

Nevertheless, ISSRs can also be converted into co-dominant markers, such as Sequence Characterized Amplified Regions, to increase their utility. Compared to other widely used marker systems, ISSRs are straightforward, less complicated technically, easier to use, and less expensive [24]. ISSR analysis, a more recent source of genetic markers, has also proved efficient in assessing genetic diversity relationships among *Olea europaea* genotypes [13,25]. The most significant advantage of ISSR markers is that they eliminate the need for costly and time-consuming genomic or other library-stage construction [26]. The ISSR approach, which is based on the random distribution of nucleotide units like 2, 3, 4, and 5 that are repeated in eukaryotic genomes, is highly repeatable and useful for determining genetic variation regardless of locus [26].

The present study aims to inform olive conservation and management. It uses ISSR molecular markers to evaluate the degree of genetic structure and diversity among Jordanian historic olive tree accessions.

## 2. Results

Table 1 lists the three ISSR primers that were considered, together with their sequences (5’–3’) and amplified products. A representative gel electrophoresis example is shown in (Figure 1). The banding patterns for the generated amplicons, essential for determining the D value, were 17 for the ISSR_807 and ISSR_810 markers and 18 for marker ISSR_825 (Table 1). The number of amplified polymorphic bands was 16 for the ISSR_807 marker and 17 for both the ISSR_810 and ISSR_825 markers. A total of 3150 data entries were assessed as either present bands (859 entries) or absent bands (2291 entries). Variable data entries were generated for various markers resulting in 1133 entries in ISSR_825, 1071 entries in ISSR_810, and 1071 entries in ISSR_807. However, this does not reflect their power to distinguish different populations.

According to Table 1, the computed D values for ISSR_807, ISSR_810, and ISSR_825 were 0.70, 0.61, and 0.83, respectively. A high value indicates that the marker can distinguish most studies’ accessions from each other. The PIC values were also determined (Table 1); the first marker (ISSR_807) had a value of 0.27, the second one (ISSR_810) had a value of 0.28, and the last marker (ISSR_825) had a value of 0.21.

According to Jaccard’s similarity index (Figure 2), the examined accessions shared relatively high similarities. The similarities in almost two-thirds of the samples ranged from 37% to 100%. The accession pair ‘J11’ and ‘J12,’ as well as A9 and A8, showed the highest similarity of 100%. A very low similarity level was detected between the two olive accessions J14 and ‘i20,’ which was just 4%.

The UPGMA method was applied utilizing the generated similarity matrix to produce a dendrogram, which visually shows the genetic links between the 63 examined olive accessions (Figure 3). Ten major clades were visible in the tree. Nine olive accessions (A4–A12) were grouped in the first clade (I). Another nine olive accessions (A13–A19 and A22–A23) were grouped in the second clade (II). Three olive accessions (i8, J8, A2) were grouped together with all four reference cultivars (M, N, O, F) in the third clade (III). Another six accessions (i6, i7, i9, i10, i11, i12) were grouped in the fourth clade (IV), while only three historic olive accessions (i1, i3, i4) were clustered in the fifth clade (V). Seven other accessions (i13–i19) were grouped in the sixth clade (VI). The biggest group clustered eleven accessions of historic olives (J2–J5, J7, J9–J12, i2, i5) in the seventh clade (VII). However, only two accessions (J1, A1) were clustered in the eighth clade (VIII). Seven historic olive accessions (i20–i24, A20–A21) were grouped in the ninth clade (IX), while only two accessions (J13, J14) were grouped in the tenth clade (X).

In Figure 4, the genetic structure analysis (with k = 2) revealed unique genetic content for the majority of the Ajloun accessions (green), while the remaining accessions from the other two locations (Irbid and Jerash) and the reference cultivars were resolved as a separate population (red). At k = 3, the analysis showed a major population from half of the olive accessions from Ajloun (blue), while the other half was shared with accessions from both Irbid and Jerash (red). In addition, a third population covering accessions from all three locations and the reference cultivars could be clearly distinguished (green) (Figure 4). Applying k = 4 restored the uniqueness of the Ajoun olive accessions as in k = 2; however, it resolved two major sub-populations: half of the accession (red) and another half (yellow). The remaining accession from the other two locations (Irbid and Jerash) and the reference cultivars were resolved as two additional sub-populations (blue and green). Similar to k = 4, applying higher k values in the analysis kept the uniqueness of the Ajoun olive accessions in two sub-populations, namely k = 5 (pink and blue), k = 6 (blue and red), and k = 7 (red and blue) (Figure 4). Moreover, these two sub-populations were almost free from any admixtures from other populations. However, two accessions from this region (A20 and A21) were found to be different from all other accessions from Ajloun for all k values (2–7); they rather were similar to Irbid populations.

Moreover, at k = 5, olive accessions from the other two locations showed both accessions with pure genetic material and others with admixtures. Nonetheless, the majority were separated into two major populations: Irbid (green) and Jerash (red). At k = 6, accessions from the last-mentioned locations were further divided into three populations: Irbid (yellow and pink) and Jerash (green). In addition, accessions with admixture genetic composition were still evident. A similar situation was repeated in k = 7 analysis, where the majority were divided into three populations: Irbid (yellow and blue) and Jerash (orange). The four reference cultivars (‘Mehras,’ ‘Nabali,’ ‘Manzanillo,’ and ‘Frantoio’) were grouped in one unique population at k = 5 (yellow), k = 6 (blue), and k = 7 (pink). However, ‘Mehras’ and ‘Nabali’ showed an almost pure genetic structure rather than the admixture genetic structure revealed for ‘Manzanillo’ and ‘Frantoio’ (Figure 4).

The mean value of alpha was low throughout the k values (<0.06), indicating that genetic structure analysis probably found clear clustering at all k values. Nonetheless, this does not indicate whether those clusters are real or stable, but rather that the investigated olive accessions are mostly assigned to one population or cluster.

By applying the Evanno method (Δk), it is possible to determine the uppermost hierarchical level of structure by calculating the rate of change in log-likelihoods and their variance between successive k values. In the first case of k = 2 to 3, a substantial improvement in log-likelihood was achieved along with low variance. On the other hand, for k = 3 to 4, only a small improvement was evident; however, variance was increased, indicating a less stable situation. Moreover, for k = 4 to 5, 6, and 7, log-likelihood kept increasing, though again with enlarged variance, which is a sign of overfitting and unstable solutions. Therefore, k = 3 would be the most acceptable run for the analysis; however, and as shown above, clearer and cleaner separations of sub-populations would be acceptable at k = 5.

The nineteen measured phenotypic traits showed high variability between the investigated eight groups (Table 2).

The canonical discriminant analysis, performed with the standardized canonical discriminant function coefficients for the 19 phenotypic parameters, showed that the first four functions accounted for 98.5% of the total variation (Table 3).

Based on standardized canonical discriminant function coefficients, the first canonical discriminate function, which accounted for 77.7%, was strongly influenced by stone shape index and fruit length. The second function (accounting for 11.8%) was found to be strongly influenced by flesh/stone ratio and flesh weight. The third and fourth canonical discriminate functions (accounting for 5.0% and 4.0%) were found to be influenced by leaf shape and seed weight, respectively.

The graphical representation of the distribution of the seven and referenced populations in the space of the two discriminate functions (Figure 5) showed a clear separation of all populations except for populations 1 and 2, where they overlapped between the tree samples. Furthermore, populations 4, 5, and 6 were localized close to each other when compared with other populations.

Principal component analysis (PCA) of phenotypic and ISSR data revealed clear differences in clustering patterns among the eight populations (Figure 5). In the PCA plots, each colored circle represents an individual tree (accession), and trees sharing the same color belong to the same group. The ISSR-based PCA showed distinct and well-separated clusters, where accessions within each color group clustered tightly together, indicating strong genetic differentiation among the eight groups. In contrast, the PCA derived from phenotypic traits showed that the accessions of each color group were located close to one another with partial overlap among groups, suggesting limited phenotypic variation. These findings indicated that ISSR markers provided higher discriminatory power compared with phenotypic traits. The Mantel test (r = 0.3125, *p* = 0.0001) (Figure 6) and Procrustes analysis (r = 0.6431, *p* = 0.0001) further confirmed a significant but moderate association between molecular and phenotypic distance matrices, reflecting partial correspondence between genetic and morphological variation.

The heatmap revealed all eight distinct clusters representing the investigated populations (Figure 7). All populations were distinguished based on phenotypic parameters (phenotype) similar to DNA marker analysis (genotype). Together, the analysis of phenotypic parameters provided a complete overview of the relationships and existing variation among the historic olive trees in this investigation. Moreover, phenotypic parameters were clustered together for similar organ-like leaves or fruits (Figure 8).

All 19 phenotypic parameters were applied to correlation analysis. Strong positive correlations were found between leaf perimeter and leaf length (99%), flesh thickness and fruit width (98%), flesh/stone ratio and flesh weight (96%). On the other hand, strong negative correlations were found between stone% and flesh% (−100%), flesh/stone ratio and stone% (−87%), and flesh thickness and fruit shape index (−82%).

## 3. Discussion

An investigation of historic olives in Jordan was the major driving force to undertake the current study. Historic olives are considered the backup for the global olive production sector, including research. Our main focus was to elucidate the genetic structure of historic olive genotypes to identify promising genetic material that could be used as a potential genetic resource to overcome major challenges of olive production. This study covered a vast majority of historic olive accessions grown for hundreds of years in the northern regions of Jordan [8,23], which is known as the origin of cultivation and utilization of olives thousands of years ago [4,5,6,7]. These living historic genotypes were in urgent need for genetic structure analysis to elucidate available populations and any admixtures as they were unintentionally excluded from pan-Mediterranean diversity studies [16,17]. A recent study revealed that the two deeply explored historic olive cultivars possess major abiotic stress-specific biomarkers, e.g., seven-fold up-regulated expression in xyloglucan endotransglucosylase hydrolase under salinity stress in both ‘Mehras’ and ‘Nabali,’ respectively [14]. In the current study, the generated dendrogram grouped ‘Mehras’ (M) and ‘Nabali’ (N) cultivars with the historic olive accessions I8, J8, and A2 (Figure 3). This was further evident in the genetic structure analysis at k = 5 (Figure 4). These promising accessions are potential candidates for salinity tolerance studies similar to ‘Nabali’ [15].

In the current study, ISSR marker techniques were used to assess a total of sixty-three historic olive accessions from the northern sector of Jordan—Irbid, Jerash, and Ajloun. The molecular markers (ISSR_807, ISSR_810, and ISSR_825) delivered relatively high scores for discriminating power (0.70, 0.61, and 0.83, respectively) (Table 1). The results of an earlier study [13] showed higher D values because they investigated diverse cultivars from around the Mediterranean basin rather than historic accessions of closely related populations. It is expected to have more genetic variation between cultivars and hence higher D values for DNA markers [12,13] than for related individuals in related populations as is the case in this study. Based on the similarity index of Jaccard (Figure 2), almost two-thirds of the investigated accessions showed from 37% to 100% similarity values. Accessions ‘J11,’ ‘J12,’ A9, and ‘A8’ exhibited the highest value of 100% similarity, meaning they are closely related and not all of them need to be investigated for potential utilization to overcome any olive production challenge, e.g., salinity and drought [23]. In contrast, the two olive accessions J14 and ‘i20’ had an extremely low similarity level (9%) and would deliver unique and promising genetic material for future studies.

The generated dendrogram along with the genetic structure analysis revealed several take-home messages for future work with these invaluable genetic resources. It is of course too expensive to investigate hundreds of thousands of historic trees in any specific study; this is hindered by time and cost. Nonetheless, the olive groves investigated in this study varied in their size and tree distribution in the field. We were keen to get multiple samples from the same grove when possible, which turned out to be more difficult in historic groves as farmers were propagating their needs of new olive plants from their own olive mother plants. Most duplicate sample accessions were similar in genetic material. Furthermore, the order of historic trees in any given grove is an indication of its historic dimension; more evenly spaced trees are expected in recently cultivated groves while sporadic trees in any grove are a sign of a much older plantation. This is simply the result of the natural spread of any given historic tree, as it continues giving new shoots around its perimeter while the central branch ages and dies. With continuous events over hundreds of years, trees are no longer evenly distributed in an olive grove but rather more sporadically. It is important to investigate older historic groves rather than new ones because of their precious genetic material that can deliver plausible solutions to current challenges and the impacts of climate change.

The second major outcome was the discovery of two major sub-populations for historic olive accessions from Ajloun region. It is important to investigate these two distinct genetic pools, along with available reference cultivars, for potential utilization in comparable stress environments (both biotic and abiotic) to disclose their potential, if any [12,13,19]. Nonetheless, remaining unique historic olive accessions with either pure or admixture genetic makeup are also worth further investigation.

High-throughput sequencing technology was used to determine the mitochondrial genome of the historic olive cultivar ‘Mehras’ by utilizing NGS [23] in addition to the chloroplast genome sequence [8]. In line with their conclusions based on plastome and mitogenome sequencing, the investigations showed that the olive cultivar ‘Mehras’ shares an ancient common ancestor with the ‘Frantoio’ cultivar from Italy and the ‘Manzanilla’ cultivar from Spain. Although ‘Manzanilla,’ ‘Mehras,’ and ‘Nabali’ as well as ‘Frantoio,’ which served as the study’s reference samples, they cluster in the same clade; ‘Frantoio’ and ‘Manzanilla’ cluster in the second group in the dendrogram of the current study. This should by no means indicate any contradictory conclusions. Genetic diversity studies for diverse cultivars from different regions around the Mediterranean are expected to amplify the diversity between them and to discriminate between them [12,13], similar to phylogenetic studies employing plastomes [8] or mitogenomes [23]. However, investigating a group of individuals belonging to related populations would surely show individuals of any out groups more related to each other, as was the case in the current study (Figure 3). Moreover, the applied genotyping technology would give different insights. For olive cultivars, three Mediterranean groups (eastern, western, and northern) were resolved using SSR markers [16], while three different Mediterranean groups (northeast, north-A, and north-B) were resolved using thousands of NGS-generated SNPs [17]. However, both studies were missing any historic or current olive cultivars from Jordan. This makes the current study a major enrichment for previous investigations. In this regard, it is worth mentioning that just one deeply investigated historic cultivar, ‘Mehras,’ was found to share a common ancestor with major European cultivars (eastern, northern, and western) [8,23].

The phenotypic characterization revealed seven promising populations of historic olives. This was evident by the ability of phenotypic parameters to distinguish all populations with the canonical discriminant analysis (Figure 5) and hierarchical clustering analysis (Figure 7). The support of molecular analysis with phenotypic characterization supports the major outcome of this study, revealing promising olive historic genotypes that can be utilized in future breeding and utilization studies.

The PCA results clearly demonstrate the superior ability of ISSR markers to discriminate among genotypes compared with phenotypic data. In the ISSR-based PCA, trees belonging to the same group (represented by the same color) (Figure 5) formed compact and distinct clusters, confirming that ISSR markers effectively capture underlying genetic diversity. This pattern reflects the high polymorphism of ISSR loci and their broad genome coverage, making them valuable for differentiating closely related genotypes. In contrast, the phenotypic PCA revealed less separation among groups, with considerable overlap between accessions of different colors. Although accessions within the same group tended to cluster together, the overall pattern suggested weaker resolution. This may be due to environmental influences, phenotypic plasticity, or the limited number of morphological traits evaluated. As a result, phenotypic variation alone was insufficient to clearly distinguish the groups identified by molecular data.

Nevertheless, the significant Mantel (Figure 6) and Procrustes correlations indicate that the two datasets share a moderate degree of correspondence. This suggests that at least part of the phenotypic variation observed among trees has a genetic basis, while other components likely reflect environmental or developmental effects. Together, these results emphasize that combining molecular and phenotypic data provides a more complete understanding of diversity patterns and relationships among the studied accessions.

The findings of this study would recommend investigating identified potential historic olive accessions with additional molecular markers and studying them under stress environments. Furthermore, it is of urgent need to multiply these unique accessions vegetative using rooted cuttings, grafting to rootstocks, or micropropagation using plant tissue culture.

## 4. Materials and Methods

Plant material was collected from the leaves of historic olive trees grown in three northern governorates of Ajloun, Jerash, and Irbid. There were 24 samples from Irbid (I), 13 from Jerash (J), and 22 from Ajloun (A) (Table 4) and (Figure 9). The National Agricultural Research Center (NARC) in Jordan’s Mshaqar research station kindly provided us with reference cultivar samples: ‘Mehras’ (M), ‘Nabali’ (N), ‘Manzanillo’ (O), and ‘Frantoia’ (F). Samples of healthy green leaves were taken from trees. Samples of leaves were delivered to the lab in an ice box and kept in the deep freezer at −80 °C.

### 4.1. DNA Extraction

Total genomic DNA was extracted from 1 cm^2^ leaf samples using the CTAB (Cetyl Trimethyl Ammonium Bromide) method [27]. DNA was rehydrated using TE buffer (Tris EDTA (ethylenediaminetetraacetic acid)) in the fridge overnight. The DNA was tested on 0.8% agarose gel and diluted with sterile distilled water to a 5 ng/µL concentration.

### 4.2. DNA Amplification

Polymerase Chain Reaction (PCR) was used to carry out the amplification reactions. ISSR markers were used to examine DNA samples. Three primers were used for ISSR analysis (Table 1). The amplification was carried out in a 20 μL reaction mixture that contained 6 μL of distilled water, 2 μL of primer (10 μM), 2 μL of extracted template DNA (5 ng/μL), and 10 μL of master mix (Trans-Gen Biotech, Beijing, China) in 200 μL PCR microtubes.

A Veriti thermal cycler (Applied Biosystems, Foster City, CA, USA) with a heated top was used for PCR. Three phases made up the PCR program: 5 min at 95 °C followed by 45 amplification cycles (95 °C for 30 s, and 45 s at 50 °C, 1 min at 72 °C) and finally an additional 10 min extension step at 72 °C. Following amplification, 1.8% agarose was employed in gel electrophoresis to separate the amplified products, and the gels were run for 150 min with 90 volts in 1× TBE buffer. A 100 base pair (bp) molecular size ladder (BIO-HELIX, New Taipei, Taiwan) was used to estimate the fragment sizes, and they were detected by using RedSafe TM Nucleic Acid Staining Solution. PCR products were visualized using an Alpha Imager gel documentation system (Alpha Imager 2200, Alpha Innotech, San Leandro, CA, USA) equipped with a UV-trans illuminator.

### 4.3. DNA Data Analysis and Interpretation

Only bright and reproducible amplified bands were assessed for analysis, and each run was repeated at least three times for verification. The scores for amplification products were absent (0) or present (1). The software program SPSS (Statistical Package for the Social Sciences) version 22 was used to compute the similarity matrix of the ISSR profile scores using Jaccard similarity indices [28]. The similarity index is equal to the number of common bands divided by the number of bands not in common. Using features available in SPSS version 22, a dendrogram was created based on the Unweighted Pair Group Method with Arithmetic Mean (UPGMA).

Based on the number of alleles found and the frequency of their distribution, a marker’s capacity to demonstrate polymorphism in the population was used to calculate its Polymorphism Information Content (PIC). The number of known (established) alleles and the frequency of their distribution determine the marker’s discriminating capacity, which is determined by PIC [29].PIC = 2ƒ (1 − ƒ)

In which ƒ = the frequency of present bands in the developing gel and 1 − ƒ = frequency of absent bands. The highest PIC value for the dominant markers is 0.5. Keep in mind that PIC values are greater for markers that are equally distributed throughout the population. The numbers are significantly greater for the markers with several alleles, but the distribution frequency of the alleles also affects the PIC value. The discriminating power (D) was computed in order to examine how well ISSR primers identified various olive genotypes [30]:$$Dj=1-C_{j}=1-\sum_{i=1}^{I}\left(p_{i}\frac{\left(Npi-1\right)}{N-1}\right)$$
where *p_i_* is the frequency of the *i*th pattern for a given *j*th primer and *Cj* (probability of confusion) is the total of the various ci for all I patterns produced by the primer for a collection of individuals (*N*).

Using Structure ver. 2.3.4 software [31], we employed a model-based clustering technique for our multi-locus microsatellite dataset to ascertain the number of clusters and the revealed genomic structure of the studied olive samples. k = 2 to k = 7 were chosen in order to estimate the number of populations (K). Assuming no prior population information, each run included 30,000 MCMC (Markov Chain Monte Carlo) repetitions after a 30,000-step burn phase using associated allele frequencies and an admixture model [32].

### 4.4. Phenotypic Data

At harvest time (11 October 2025), fruits and leaves were collected from a group of five samples for the eight cluster groups resolved by the DNA dendrogram (Figure 3) and gene structure analysis (Figure 4), where cluster number for DNA (I, II, IV, V, VI, VII, IX, III) corresponded to phenotypic data numbers (1, 2, 3, 4, 5, 6, 7, 8), and clusters VIII and X were excluded as they represented just two accessions and were not well resolved in gene structure analysis (Figure 4). Leaf measured parameters were as follows: leaf length, leaf width, leaf area, leaf shape index (leaf length divided by leaf width), leaf perimeter (based on Ramanujan’s equation [33]), and leaf roundness (based on Russ’s equation [34]. Fruit measured parameters were as follows: fruit length, fruit width, fruit shape index (fruit length divided by fruit width), and fruit weight. Fruit stones were removed with an olive pitter and exposed to 5% for 15 min and washed thoroughly with water before drying with towel paper. The stone measured parameters were as follows: stone length, stone width, stone shape index (stone length divided by stone width), stone weight, and stone percentage (stone weight divided by fruit weight). Finally, the fruit flesh measurements were as follows: flesh thickness, flesh weight, flesh percentage (flesh weight divided by fruit weight) and flesh/stone ratio (weight by weight).

The measured data was utilized for analysis. Canonical discriminant analysis (CDA) was performed using SPSS 18. Moreover, hierarchical clustering based on Manhattan distance was performed to genrate a clusters heatmap for both phenotypic parameters and investigated samples. In addition, correlation was assessed between all 19 phenotypic parameters.

### 4.5. Correlation Between Phenotypic and ISSR Data

Phenotypic and ISSR marker data were analyzed to assess the relationships among the studied historic olive accessions. Two datasets (phenotypic traits and ISSR binary matrix) were imported and processed in R version 4.4.0 (https://cran.r-project.org/bin/windows/base/old/4.4.0/, accessed on 1 November 2025). Data handling and visualization were performed using the base R environment and the packages vegan [35], phangorn [36], dendextend [37], and ggplot2 [38], all accessed on 1 November 2025.

Pairwise distance matrices were computed using Euclidean distance for phenotypic traits and Jaccard distance for ISSR binary data. Cluster analyses were carried out using the UPGMA (Unweighted Pair Group Method with Arithmetic Mean) algorithm. The correlation between molecular and phenotypic distance matrices was tested using the Mantel test with 9999 permutations [39]. A Procrustes analysis and the associated Protest test were applied to assess the similarity between the ordination spaces of both datasets. Principal component analyses (PCA) were also conducted separately for phenotypic and ISSR data to visualize grouping patterns among accessions.

## 5. Conclusions

In summary, our study demonstrated that utilizing ISSR markers and phenotypic data as powerful tools are essential for determining the genetic makeup of promising historic olive accessions, providing precise characterizations and opening the door for valorization of these genetic resources.

## Figures and Tables

**Figure 1 ijms-26-10863-f001:**
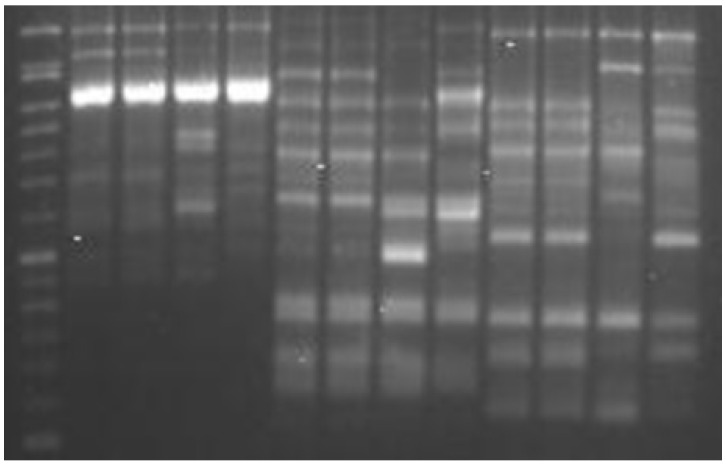
Gel electrophoresis of sample of investigated olive accessions. Left lane indicates a 100 bp ladder marker.

**Figure 2 ijms-26-10863-f002:**
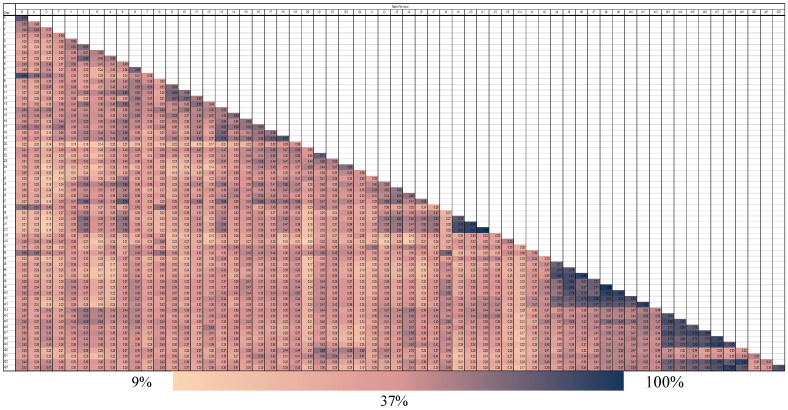
Jaccard’s similarity index generated for olive cultivars as obtained from ISSR data.

**Figure 3 ijms-26-10863-f003:**
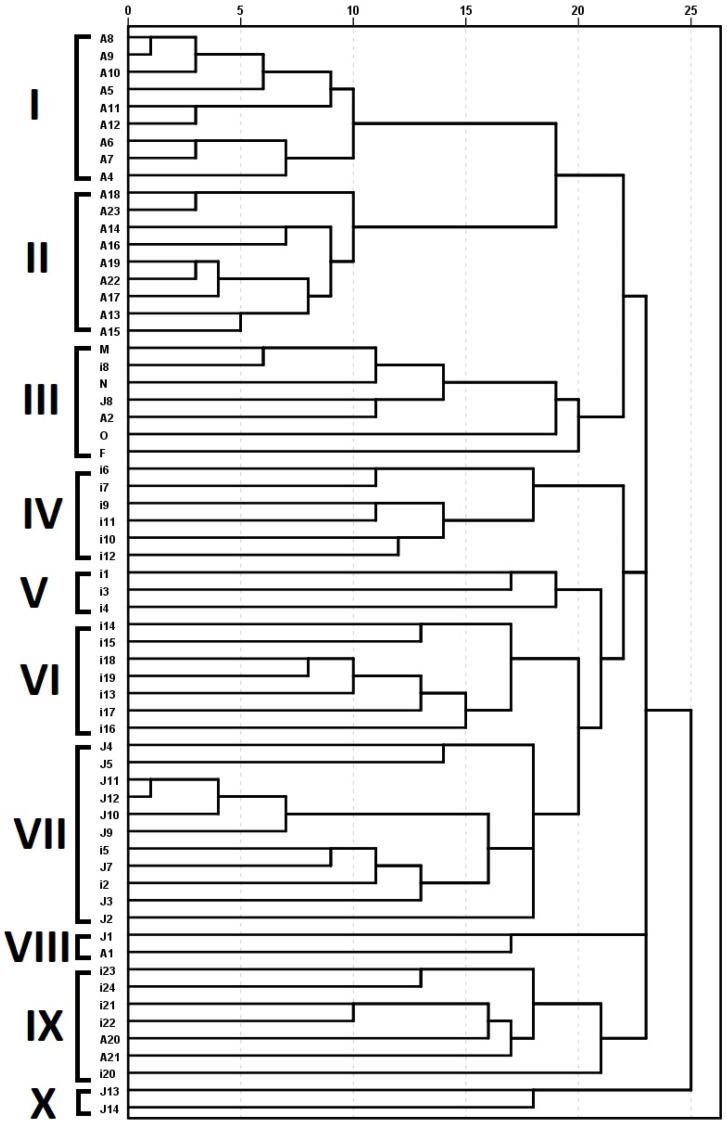
Dendogram of olive cultivars generated by UPGMA cluster analysis of the genetic similarity values based on ISSR data.

**Figure 4 ijms-26-10863-f004:**
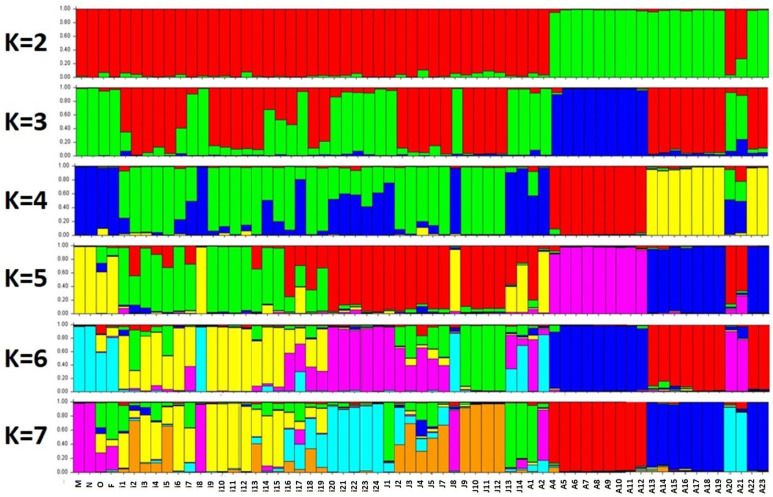
Genetic structure analysis of historic olive accession along four reference cultivars (M, N, O, F), as inferred at k = 2 to 7. Each accession is represented by a vertical bar divided into colored segments (for admixtures), the lengths of which indicate the proportions of the genome attributed to the inferred accession.

**Figure 5 ijms-26-10863-f005:**
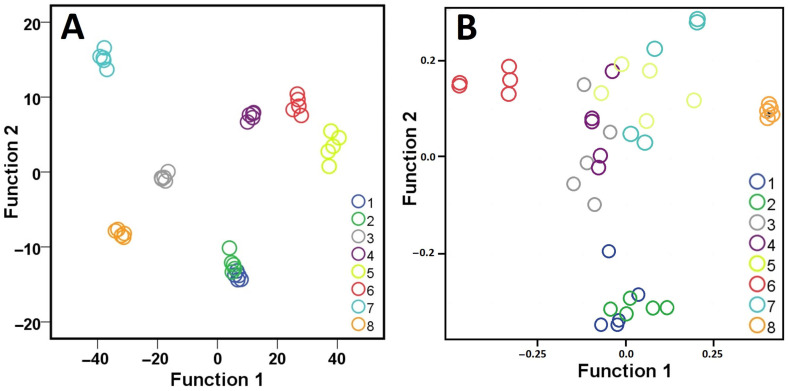
(**A**) Canonical discriminant functions between the eight Jordanian populations of historic olives (1–8) from clusters analysis (see Materials and Methods Section for more details) and the two main functions, performed on the basis of 19 phenotypic parameters. (**B**) Similar PCA for the ISSR molecular data for the same samples (five each).

**Figure 6 ijms-26-10863-f006:**
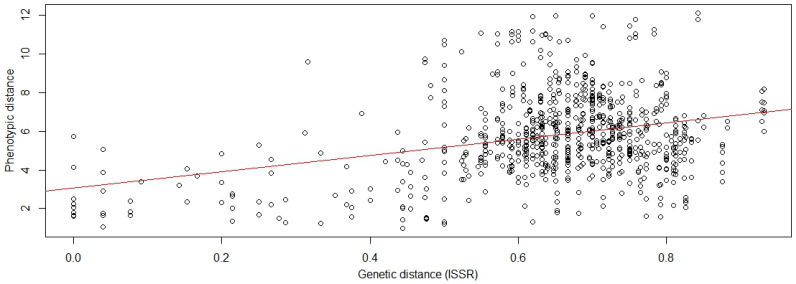
Mantel correlation between the phenotypic distances and genetic distances, r = 0.312.

**Figure 7 ijms-26-10863-f007:**
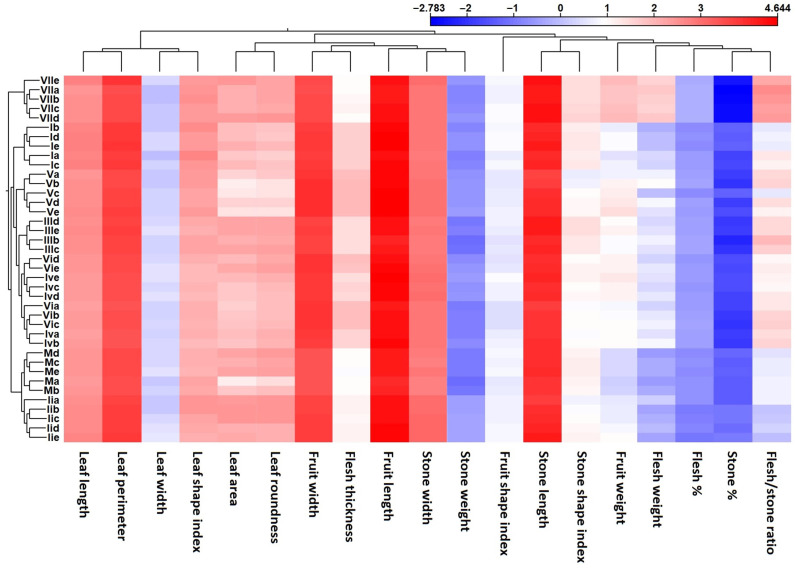
Heatmap and two-dimensional hierarchical clustering using data from 19 phenotypic parameters measured in the 40 olive tree samples under study.

**Figure 8 ijms-26-10863-f008:**
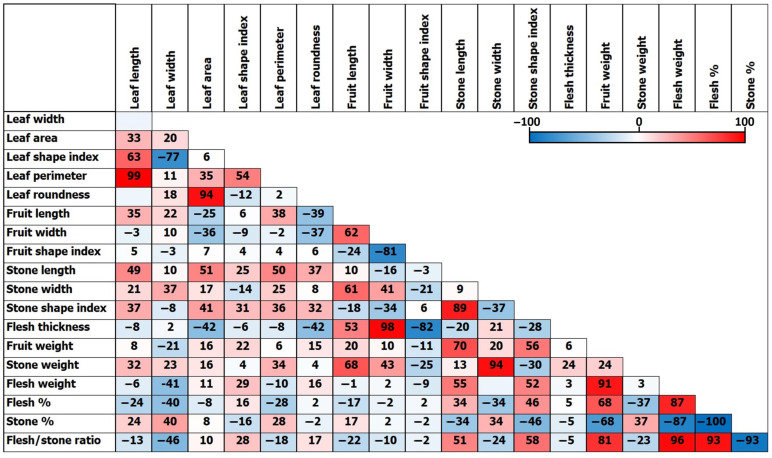
Correlations (%) between 19 phenotypic parameters along all studied olive tree samples.

**Figure 9 ijms-26-10863-f009:**
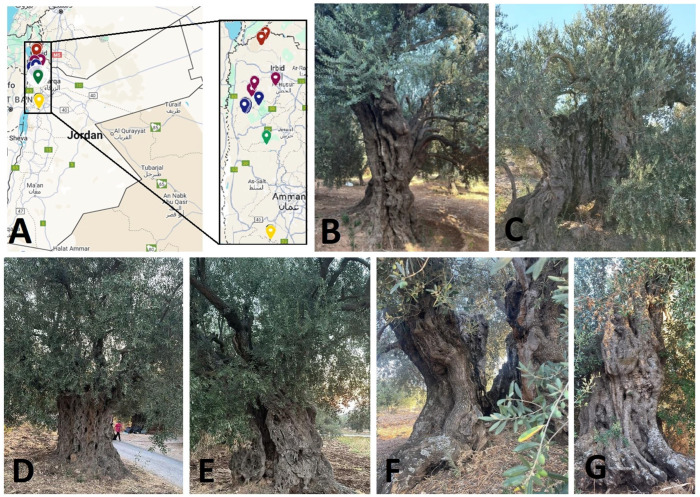
(**A**) Jordan map with zoom-in showing the distribution of collected accessions from three northern governorates. Examples of olive accessions: (**B**) J8, (**C**) i18, (**D**) A2, (**E**) A10, (**F**) i5, (**G**) i17. Photos are courtesy of N. Alsakarneh.

**Table 1 ijms-26-10863-t001:** ISSR DNA primers along with their sequences, number of banding patterns, discriminating power (D), and total and polymorphic bands generated that are resultant from all tested olive genotypes.

Primer	Sequences (5′–3′)	Patterns *	D Value	Pic Value	Amplified Fragments	
					Total	Polymorphic
ISSR_807	AGAGAGAGAGAGAGAGT	63	0.70	0.27	17	16
ISSR_810	GAGAGAGAGAGAGAGAT	63	0.61	0.28	17	17
ISSR_825	ACACACACACACACACT	63	0.83	0.21	18	17

* Needed to calculate the D value.

**Table 2 ijms-26-10863-t002:** The mean values (M) and standard deviation (SD) for nineteen phenotypic parameters for the eight historic olive populations (each with five accessions).

	Group															
Parameter	1		2		3		4		5		6		7		8	
	M	SD	M	SD	M	SD	M	SD	M	SD	M	SD	M	SD	M	SD
Leaf length (cm)	6.46	0.11	6.04	0.18	5.76	0.09	5.08	0.13	5.70	0.19	5.18	0.19	5.82	0.51	5.36	0.11
Leaf width (cm)	1.18	0.13	1.28	0.18	1.26	0.09	1.32	0.08	1.16	0.05	1.28	0.08	1.10	0.12	1.26	0.11
Leaf area (cm^2^)	3.66	0.37	4.90	0.54	4.45	0.26	3.43	0.21	2.58	0.28	3.58	0.59	4.60	0.55	3.82	1.04
Leaf shape index	5.52	0.54	4.78	0.56	4.59	0.24	3.86	0.16	4.92	0.16	4.05	0.14	5.30	0.19	4.28	0.31
Leaf perimeter (cm)	13.49	0.32	12.76	0.49	12.20	0.25	10.94	0.33	11.99	0.40	11.10	0.44	12.18	1.08	11.43	0.32
Leaf roundness	3.41	0.28	4.85	0.68	4.59	0.21	3.94	0.29	2.71	0.33	4.04	0.51	4.74	0.35	4.19	1.05
Fruit length (mm)	23.00	1.22	21.20	0.84	18.40	1.52	22.40	1.14	23.60	1.14	20.40	1.14	20.00	1.41	17.80	1.30
Fruit width (mm)	13.32	0.07	12.56	0.21	10.15	4.87	13.41	0.11	15.06	0.24	14.59	0.09	11.43	0.32	10.67	0.12
Fruit shape index	1.73	0.09	1.69	0.04	3.56	4.56	1.67	0.08	1.57	0.05	1.40	0.07	1.75	0.10	1.67	0.11
Stone length (mm)	16.80	0.84	15.80	1.92	17.00	1.58	15.40	1.14	14.40	1.82	14.40	1.14	20.00	1.00	14.80	0.84
Stone width (mm)	7.19	0.11	8.29	0.27	6.95	0.19	7.73	0.26	7.67	0.08	7.39	0.21	7.38	0.33	6.83	0.19
Stone shape index	2.34	0.08	1.90	0.18	2.44	0.17	1.99	0.08	1.88	0.22	1.95	0.10	2.71	0.09	2.17	0.07
Flesh thickness (mm)	3.07	0.03	2.14	0.06	1.60	2.38	2.84	0.10	3.70	0.09	3.60	0.07	2.03	0.05	1.92	0.07
Fruit weight (g)	1.65	0.17	1.66	0.15	1.83	0.24	2.11	0.16	2.21	0.37	1.99	0.19	3.66	0.19	1.25	0.08
Stone weight (g)	0.58	0.04	0.76	0.03	0.47	0.04	0.62	0.05	0.65	0.02	0.57	0.06	0.60	0.04	0.49	0.03
Flesh weight (g)	1.07	0.18	0.90	0.17	1.37	0.23	1.49	0.15	1.56	0.37	1.42	0.18	3.06	0.20	0.76	0.06
Flesh %	0.64	0.05	0.54	0.05	0.74	0.03	0.70	0.03	0.70	0.06	0.71	0.03	0.84	0.01	0.61	0.01
Stone %	0.36	0.05	0.46	0.05	0.26	0.03	0.30	0.03	0.30	0.06	0.29	0.03	0.16	0.01	0.39	0.01
Flesh/stone ratio	1.86	0.41	1.20	0.27	2.93	0.52	2.41	0.33	2.40	0.59	2.51	0.40	5.11	0.55	1.57	0.10

**Table 3 ijms-26-10863-t003:** Eigenvalues and percent of variability explained by each canonical discriminant function for the eight Jordanian populations of historic olives, including the reference ‘Mehras’ (number 8) references populations based on 19 phenotypic parameters.

Function	Eigenvalue	% of Variance	Cumulative %
1	809.156	77.7	77.7
2	122.815	11.8	89.5
3	51.861	5.0	94.5
4	41.699	4.0	98.5

**Table 4 ijms-26-10863-t004:** Historic olive accessions along with collection region, area and GPS.

Acc.	Region	Area	GPS	Acc.	Region	Area	GPS
M	Mshaqar	Mehras	31.77394, 35.80324	J5	Jerash	Borma	32.21637, 35.77961
N	Mshaqar	Nabali	31.77394, 35.80324	J7	Jerash	Borma	32.21637, 35.77961
O	Mshaqar	Manzanillo	31.77394, 35.80324	J8	Jerash	Borma	32.21637, 35.77961
F	Mshaqar	Frantoio	31.77394, 35.80324	J9	Jerash	Borma	32.21637, 35.77961
i1	Irbid	Saham al Kaffarat	32.70001, 35.77391	J10	Jerash	Borma	32.21637, 35.77961
i2	Irbid	Saham al Kaffarat	32.70001, 35.77391	J11	Jerash	Borma	32.21637, 35.77961
i3	Irbid	Saham al Kaffarat	32.70001, 35.77391	J12	Jerash	Borma	32.21637, 35.77961
i4	Irbid	Saham al Kaffarat	32.70001, 35.77391	J13	Jerash	Borma	32.21637, 35.77961
i5	Irbid	Saham al Kaffarat	32.70001, 35.77391	J14	Jerash	Borma	32.21637, 35.77961
i6	Irbid	Saham al Kaffarat	32.70001, 35.77391	A1	Ajloun	Al Hashimiyya (Al mesier)	32.36958, 35.65913
i7	Irbid	Saham al Kaffarat	32.70001, 35.77391	A2	Ajloun	Al Hashimiyya (Al mesier)	32.36958, 35.65913
i8	Irbid	Saham al Kaffarat	32.70001, 35.77391	A4	Ajloun	Al Hashimiyya (Al mesier)	32.36958, 35.65913
i9	Irbid	Saham al Kaffarat	32.70001, 35.77391	A5	Ajloun	Al Hashimiyya (Al mesier)	32.36958, 35.65913
i10	Irbid	Saham al Kaffarat	32.70001, 35.77391	A6	Ajloun	Al Hashimiyya (Al mesier)	32.36958, 35.65913
i11	Irbid	Saham al Kaffarat	32.70001, 35.77391	A7	Ajloun	Al Hashimiyya (Al mesier)	32.36958, 35.65913
i12	Irbid	Saham al Kaffarat	32.70001, 35.77391	A8	Ajloun	Al Hashimiyya (Al mesier)	32.36958, 35.65913
i13	Irbid	Saham al Kaffarat	32.70001, 35.77391	A9	Ajloun	Al Hashimiyya (Al mesier)	32.36958, 35.65913
i14	Irbid	Saham al Kaffarat	32.70001, 35.77391	A10	Ajloun	Al Hashimiyya (Al mesier)	32.36958, 35.65913
i15	Irbid	Saham al Kaffarat	32.70001, 35.77391	A11	Ajloun	Al Hashimiyya (Ashlaa)	32.36147, 35.66961
i16	Irbid	Saham al Kaffarat	32.70001, 35.77391	A12	Ajloun	Al Hashimiyya (Ashlaa)	32.36147, 35.66961
i17	Irbid	Saham al Kaffarat	32.70001, 35.77391	A13	Ajloun	Al Hashimiyya (Ashlaa)	32.36147, 35.66961
i18	Irbid	Saham al Kaffarat	32.70001, 35.77391	A14	Ajloun	Al Hashimiyya (Ashlaa)	32.36147, 35.66961
i19	Irbid	Al-Korah	32.44289, 35.70047	A15	Ajloun	Al Hashimiyya (Ashlaa)	32.36147, 35.66961
i20	Irbid	Al-Korah	32.43796, 35.69473	A16	Ajloun	Al Hashimiyya (Ashlaa)	32.36147, 35.66961
i21	Irbid	Malka	32.67751, 35.74889	A17	Ajloun	Orjan	32.36291, 35.65904
i22	Irbid	Malka	32.67589, 35.75027	A18	Ajloun	Al Hashimiyya (Amesier)	32.36291, 35.65904
i23	Irbid	Der Abo Yousef	32.48906, 35.83198	A19	Ajloun	Al Hashimiyya (Amesier)	32.36207, 35.65798
i24	Irbid	Al-Korah	32.46954, 35.71007	A20	Ajloun	Al Hashimiyya (Qarmash)	32.36207, 35.65798
J1	Jerash	Borma	32.21433, 35.78064	A21	Ajloun	Ajloun	32.36432, 35.65283
J2	Jerash	Borma	32.21433, 35.78064	A22	Ajloun	Wadi El Rayan	32.39776, 35.73709
J3	Jerash	Borma	32.21637, 35.77961	A23	Ajloun	Wadi El Rayan	32.39776, 35.73709
J4	Jerash	Borma	32.21637, 35.77961				

## Data Availability

The original contributions presented in this study are included in the article. Further inquiries can be directed to the corresponding authors.

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
