# Peer review of "Population Genetic Structure of Historic Olives (Olea europaea subsp. europaea) from Jordan"

_ijms, 2025, doi:10.3390/ijms262210863_

Round 1

Reviewer 1 Report (Previous Reviewer 3)

Comments and Suggestions for Authors

Dear Dr.,

the present article 'Population genetic structure of historic olives (Olea europaea subsp.europaea) from Jordan' is really difficult to assess. Authors have incorporated phenotypic data, but now they haven´t integrated well in the manuscript. They have worked the phenotypic data too fast, without relating them with the genotypic ones. To my opinion they must improve the following aspects:

  • Regarding genotypic data, Figure 3 and Figure 4 have contradicted information. The former gives 10 groups, while the latter 3-7 groups.
  • Figure 5. I don´t know whether you are correlating genotypic and phenotypic data in this figure. I would like to see the correlation in a clearer way.
  • Where are the phenotypic data (at least the mean) of the eight population of Jordanian olive trees. I don´t see them anywhere.
  • Discussion: the discussion of the phenotypic data is just a paragraph, with little value. Authors say that phenotypically they found seven 'novel' population of historic olives. First, the groups are not 'novel', second, how do you reconcile this seven phenotypic groups with the eight genotypic group? I see that one genotypic group is loosed.

The authors need to join quietly and to reflect on the goals of the article. The rush that I see in the writing of this article (especially in the review) does not help to improve the quality of the manuscript.

Best

Author Response

Response to Reviewer 1:

Dear Dr.,

the present article 'Population genetic structure of historic olives (Olea europaea subsp.europaea) from Jordan' is really difficult to assess. Authors have incorporated phenotypic data, but now they haven´t integrated well in the manuscript. They have worked the phenotypic data too fast, without relating them with the genotypic ones. To my opinion they must improve the following aspects:

  • Highly appreciate the valuable comments of the respected reviewer. Yes you are write, therefore, a Phenotypic and ISSR data were analyzed using PCA, cluster, and Mantel tests to examine the relationship between the phenotypic data and the ISSR data. Additional PCA for molecular data was added and Mantel plot was added with related text in methodology, results and discussion sections.

  • Regarding genotypic data, Figure 3 and Figure 4 have contradicted information. The former gives 10 groups, while the latter 3-7 groups.
  •  
  • Thank you for this important difference. Cluster number for DNA (I, II, IV, V, VI, VII, IX, III) correspond to phenotypic data numbers (1, 2, 3, 4, 5, 6, 7, 8), and clusters VIII and X were excluded as it represent just two accessions and were not well resolved in gene structure analysis. This was added to the methodology section
  •  
  • Figure 5. I don´t know whether you are correlating genotypic and phenotypic data in this figure. I would like to see the correlation in a clearer way.
  •  
  • Sure this was done in the revised manuscript with PCA and Mantel test. with related text in methodology, results and discussion sections.

  • Where are the phenotypic data (at least the mean) of the eight population of Jordanian olive trees. I don´t see them anywhere.
  •  
  • Thank you for this important comment. A new table (2) was added with mean values and SD for all 19 phenotypic data covering the eight populations.
  •  
  • Discussion: the discussion of the phenotypic data is just a paragraph, with little value. Authors say that phenotypically they found seven 'novel' population of historic olives. First, the groups are not 'novel', second, how do you reconcile this seven phenotypic groups with the eight genotypic group? I see that one genotypic group is loosed.
  •  
  • Thank you for this comment. Three additional paragraphs were added in the discussion section covering the relationship between phenotypic data and ISSR data.

The authors need to join quietly and to reflect on the goals of the article. The rush that I see in the writing of this article (especially in the review) does not help to improve the quality of the manuscript.

  • Thank you for this comment. The revised manuscript covers all the requested suggestions and comment, hoping to be much better.

Reviewer 2 Report (New Reviewer)

Comments and Suggestions for Authors

This study presents a valuable first systematic analysis of the genetic diversity and structure of historic olive germplasm from northern Jordan, a region underrepresented in Mediterranean olive genetic studies. The combination of ISSR markers and phenotypic characterization provides complementary evidence. The research addresses an important gap, but several aspects require strengthening to enhance its scientific rigor and impact.

1. Sample Strategy and Representation

Comment: The sampling is concentrated in three northern governorates, with multiple accessions collected from the same orchard (e.g., Borma in Jerash). This may introduce redundancy and potentially bias the diversity analysis by overrepresenting certain genotypes.

2. Limited Number of Molecular Markers

Comment: The use of only three ISSR primers, while yielding polymorphic bands, may not provide sufficient genome coverage to capture the full extent of genetic diversity. This limits the robustness and comparability with other studies that often employ larger marker sets (e.g., SSRs or SNPs).

3. Population Genetic Structure Analysis

Comment: The interpretation of the STRUCTURE analysis is somewhat confusing, particularly the rationale for discussing K=5 when the Evanno method suggests K=3 is optimal. The description of the results for different K-values is text-heavy and difficult to follow.

The manuscript requires thorough proofreading to improve fluency and conciseness. Some sentences are repetitive (e.g., multiple descriptions of the 'Mehras' cultivar's significance), and others are awkwardly constructed. Seek assistance from a native English speaker or professional editing service to enhance readability and ensure grammatical accuracy.

This manuscript presents a valuable contribution to the field. After the authors have addressed the major concerns (particularly regarding the interpretation of population structure and the acknowledgment of methodological limitations) and the minor points listed above, I recommend Minor Revisions for potential publication in the International Journal of Molecular Sciences.

Author Response

This study presents a valuable first systematic analysis of the genetic diversity and structure of historic olive germplasm from northern Jordan, a region underrepresented in Mediterranean olive genetic studies. The combination of ISSR markers and phenotypic characterization provides complementary evidence. The research addresses an important gap, but several aspects require strengthening to enhance its scientific rigor and impact.

  • We would like to thank the respected reviewer for this comment.

  1. Sample Strategy and Representation

Comment: The sampling is concentrated in three northern governorates, with multiple accessions collected from the same orchard (e.g., Borma in Jerash). This may introduce redundancy and potentially bias the diversity analysis by overrepresenting certain genotypes.

  • Thank is very importan6t comment. We did not know the real gene structure of any of the investigated populations until we performed this study, which is an important one for future sampling, breeding, and conservation.

  1. Limited Number of Molecular Markers

Comment: The use of only three ISSR primers, while yielding polymorphic bands, may not provide sufficient genome coverage to capture the full extent of genetic diversity. This limits the robustness and comparability with other studies that often employ larger marker sets (e.g., SSRs or SNPs).

  • Very important comment, and it was also asked in the earlier review round to add the phenotypic data to complement the analysis. In the current revised version (R3), phenotypic and ISSR data were analyzed using PCA, cluster analysis, and Mantel tests to examine the relationship between the phenotypic data and the ISSR data. Additional PCA for molecular data was added, and Mantel plot was added with related text in the methodology, results, and discussion sections. ISSR markers showed clear genetic differentiation among groups, whereas phenotypic traits displayed lower variation but a significant correlation with molecular diversity.

  1. Population Genetic Structure Analysis

Comment: The interpretation of the STRUCTURE analysis is somewhat confusing, particularly the rationale for discussing K=5 when the Evanno method suggests K=3 is optimal. The description of the results for different K-values is text-heavy and difficult to follow.

  • You are right, however, we find it important to stay this elaborate way for any future utilization of the data.

The manuscript requires thorough proofreading to improve fluency and conciseness. Some sentences are repetitive (e.g., multiple descriptions of the 'Mehras' cultivar's significance), and others are awkwardly constructed. Seek assistance from a native English speaker or professional editing service to enhance readability and ensure grammatical accuracy.

  • This was done again for the entire manuscript.

This manuscript presents a valuable contribution to the field. After the authors have addressed the major concerns (particularly regarding the interpretation of population structure and the acknowledgment of methodological limitations) and the minor points listed above, I recommend Minor Revisions for potential publication in the International Journal of Molecular Sciences.

  • Thank you for this comment. The revised manuscript addresses all the requested suggestions and comments, with the hope of being significantly improved.

Round 2

Reviewer 1 Report (Previous Reviewer 3)

Comments and Suggestions for Authors

Dear sir,

regarding the article 'Population genetic structure of historic olives (Olea europaea subsp.europaea) from Jordan', the article is now more round and with more sense. However, I have some objections:

  • In the added text there are several spelling mistakes. I tried to correct part of them in the annotated file that I am attaching.
  • I cannot accept and believe that authors has found out 'seven novel populations of historic olives (based on phenotypic characterization)'. My guess is that they are not novel, but related to neighbouring groups of the Levant. In fact, authors haven´t checked their cultivars against a big collection of olive tree (genotyping and phenotyping), like they should do in case they wanted to proof the novelty of the Jordanian olive tree varieties. Authors could more humbly saying that... they are going to characterize the genetic structure of a collection of Jordanian varieties.
  • Therefore, if authors delete the part of the Jordanian novel varieties, I may accept the article, because most likely this is a false assumption.

Best regards

Author Response

Dear Dr.,

regarding the article 'Population genetic structure of historic olives (Olea europaea subsp.europaea) from Jordan', the article is now more round and with more sense. However, I have some objections:

  • Highly appreciate the valuable and positive comments of the respected reviewer.

In the added text there are several spelling mistakes. I tried to correct part of them in the annotated file that I am attaching.

  • Highly appreciate the valuable comments. All changes were made as in the annotated file. We high appreciate your efforts.

I cannot accept and believe that authors has found out 'seven novel populations of historic olives (based on phenotypic characterization)'. My guess is that they are not novel, but related to neighbouring groups of the Levant. In fact, authors haven´t checked their cultivars against a big collection of olive tree (genotyping and phenotyping), like they should do in case they wanted to proof the novelty of the Jordanian olive tree varieties. Authors could more humbly saying that... they are going to characterize the genetic structure of a collection of Jordanian varieties.

Therefore, if authors delete the part of the Jordanian novel varieties, I may accept the article, because most likely this is a false assumption.

  • Highly appreciate the valuable comments. And you are right; they are still NOT novel for the current study.
  • Therefore, we have deleted ALL “novel” words and replaced them with “promising” or potential” with some modification in the sentences.
  • And we would like to have your acceptance as you mentioned.
  • Again we highly appreciate your efforts along the reviewing process to reach this level of improving the manuscript.

Round 3

Reviewer 1 Report (Previous Reviewer 3)

Comments and Suggestions for Authors

Dear sir, the paper is now almost ready for publication. I have now only four comments:

  • Line 38. Put 'accessions' in plural.
  • Table 1. Put all number with the same type of letter.
  • Figure 1 is not referred in the text. Please do it.
  • Remove Figure 2. I cannot see the numbers of this table. Besides, it is not referred in the text. Therefore, you have to name the following Figures accordingly.

Best

Author Response

Dear sir, the paper is now almost ready for publication. I have now only four comments:

  • Highly appreciate the valuable and positive comments of the respected reviewer. And highly appreciate your effots.

Line 38. Put 'accessions' in plural.

Thank you, it was corrected.

Table 1. Put all number with the same type of letter.

Figure 1 is not referred in the text. Please do it.

Remove Figure 2. I cannot see the numbers of this table. Besides, it is not referred in the text. Therefore, you have to name the following Figures accordingly.

  • Highly appreciate the valuable comments, numbers of all figures (1-9) and tables (1-4) were corrected and cited in the text.

This manuscript is a resubmission of an earlier submission. The following is a list of the peer review reports and author responses from that submission.

Round 1

Reviewer 1 Report

Comments and Suggestions for Authors

The study lacks a clear statement of novelty. Given that many research has been conducted on  olive cultivars by ISSR data. The authors must explicitly state what makes this study unique and how it advances existing knowledge. The introduction should highlight specific gaps in the literature that this study aims to fill. Introduction Needs More Depth: The introduction should be expanded to include a more comprehensive discussion of population genetic structure in olives. The introduction should be enriched with new literature 
- Experimental design should be carried out and take phenotpic data and then comperd with genetic data by mantel test. Too more genetic analyzes must be performed to reach more accurate results
 Results and discussion need more depth after add  more genetic analyzes. The discussion should better relate findings to previous studies and theoretical frameworks.

Author Response

The authors would like to thank the reviewer for the constructive suggestions and the important comments, which were taken into consideration to improve the manuscript as shown in the revised version in red.

Quality of English Language

 (x) The English is fine and does not require any improvement.

  • Thank you very much for your positive comment concerning the English language. Nonetheless, the entire manuscript was critically checked for language quality and few changes were made.

Does the introduction provide sufficient background and include all relevant references? Must be improved:

  • Additional background was added as requested.

Is the research design appropriate? Must be improved:

  • Concerning the phenotypic data. We mentioned in the revised manuscript that these are historic too old trees without any ancestry data. These trees comprise around two thirds of grown olives in our region and they represent ancient uncovered potential for the future. Therefore, the first step was to scan for their genetic structure; are they the same or they differ genetically, therefore a large sample was investigated. The attained results of the current study will be applied to the next phase study, where phenotypic data will be assessed including oil quality and characteristics.

Are the methods adequately described? Must be improved

  • The methodology was further clarified as requested.

Are the results clearly presented? Must be improved

  • The presented both UPGMA analysis and dendrogram besides the genetic structure represents the most one can get from DNA marker analysis. The requested additional genetic analysis would include deep sequencing for the whole genome, this would be done at the third phase, when we will identify unique novel genotypes with unique phenotypic oil traits in the future.

Are the conclusions supported by the results? Must be improved

  • Was improved as requested.

Are all figures and tables clear and well-presented? Must be improved

  • An additional gel photo was added as an example of the generated data.

The study lacks a clear statement of novelty. Given that many research has been conducted on  olive cultivars by ISSR data. The authors must explicitly state what makes this study unique and how it advances existing knowledge. The introduction should highlight specific gaps in the literature that this study aims to fill. Introduction Needs More Depth: The introduction should be expanded to include a more comprehensive discussion of population genetic structure in olives. The introduction should be enriched with new literature:

  • The introduction was improved adding requested additional literature and covering the research gap upon which the study was carried out.

Experimental design should be carried out and take phenotypic data and then compared with genetic data by mantel test.

  • Concerning the phenotypic data. We mentioned in the revised manuscript that these are historic too old trees without any ancestry data. These trees comprise around two thirds of grown olives in our region and they represent ancient uncovered potential for the future. Therefore, the first step was to scan for their genetic structure; are they the same or they differ genetically, therefore a large sample was investigated. The attained results of the current study will be applied to the next phase study, where phenotypic data will be assessed including oil quality and characteristics.

Too more genetic analyzes must be performed to reach more accurate results.  Results and discussion need more depth after add  more genetic analyzes:

  • The presented both UPGMA analysis and dendrogram besides the genetic structure represents the most one can get from DNA marker analysis. The requested additional genetic analysis would include deep sequencing for the whole genome, this would be done at the third phase, when we will identify unique novel genotypes with unique phenotypic oil traits in the future.

The discussion should better relate findings to previous studies and theoretical frameworks:

The requested modifications were applied to the discussion. 

Reviewer 2 Report

Comments and Suggestions for Authors

The authors Alsakarneh et al. aim to elucidate the genetic structure of historic olive genotypes in order to identify novel genetic resources that can help address major challenges in olive production. This study encompasses a large number of historic olive accessions grown for centuries in northern Jordan, a region recognized as an ancient center of olive cultivation and use. The authors state that the PCR gel was run for 150 minutes, which would typically require low voltage; however, no voltage details are provided in the manuscript. Additionally, no representative gel image of the PCR amplification is included. Including at least one gel photo would enhance the reliability of the data. If available, such images should be included in the main text or as supplementary figures. 

Comments on the Quality of English Language

As a minor issue, typographical errors such as “whoch” instead of “which” and “the the” should be corrected.

Author Response

The authors would like to thank the reviewer for the constructive suggestions and the important comments, which were taken into consideration to improve the manuscript as shown in the revised version in red.

Quality of English Language

(x) The English could be improved to more clearly express the research.

  • Thank you very much for your comment, the entire manuscript was critically checked for language quality and few changes were made.

Does the introduction provide sufficient background and include all relevant references? Yes:

  • Thank you very much for positive comment.

Is the research design appropriate? Yes:

  • Thank you very much for positive comment.

Are the methods adequately described? Yes:

  • Thank you very much for positive comment.

Are the results clearly presented? Yes:

  • Thank you very much for positive comment.

Are the conclusions supported by the results? Yes:

  • Thank you very much for positive comment.

Are all figures and tables clear and well-presented? Can be improved

  • As requested a representative gel was added for generated data.

The authors Alsakarneh et al. aim to elucidate the genetic structure of historic olive genotypes in order to identify novel genetic resources that can help address major challenges in olive production. This study encompasses a large number of historic olive accessions grown for centuries in northern Jordan, a region recognized as an ancient center of olive cultivation and use.

  • Thank you for the positive comment.

The authors state that the PCR gel was run for 150 minutes, which would typically require low voltage; however, no voltage details are provided in the manuscript.

  • Requested voltage data were added.

Additionally, no representative gel image of the PCR amplification is included. Including at least one gel photo would enhance the reliability of the data. If available, such images should be included in the main text or as supplementary figures.

  • As requested one representative gel was added to the manuscript. 

Comments on the Quality of English Language

As a minor issue, typographical errors such as “whoch” instead of “which” and “the the” should be corrected.

  • Thank you very much for your comment, the entire manuscript was critically checked for language quality and few changes were made.

Reviewer 3 Report

Comments and Suggestions for Authors

Dear sir,

the present article 'Population genetic structure of historic olives (Olea europaea subsp. europaea) from Jordan' deals with a genetic structure analysis of 63 olive tree landraces from Jordan, with some reference cultivars. They seem to achieve some results regarding genetic structure of the olive material (K=2 and K=3 seems the better to me).

I think you could link your results with those presented in wider landrace sampling in the Mediterranean Basin (see Olive domestication and diversification... by Diez et al., 2015). There are even more modern literature on the subject. It seems in general that all Med. landraces may be classified in three genetic groups (east, the first; west, and central). 

Since you are presenting results of the accessions of a relatively small country, I think the relation genotype-phenotype would be interesting to present in this study. For example, fruit size, oil content, maturity date, flowering date, etc. I would consider a better accomplised study if you did so.

Minor mistakes: l. 169, insert 'a' between addition and third. l. 274 , ... the Mediterranean are expected...'. l. 439 'Olea europea' in italics.

I would consider this document to be published in this journal if the authors introduce the phenotypic data of the landraces, and try to relate genotype and phenotype.

Best

Comments on the Quality of English Language

In general, the English of this study is good.

Author Response

The authors would like to thank the reviewer (1) for the constructive suggestions and the important comments, which were taken into consideration to improve the manuscript as shown in the revised version in red.

Quality of English Language

(x) The English could be improved to more clearly express the research.

  • Thank you very much for your comment, the entire manuscript was critically checked for language quality and few changes were made.

Does the introduction provide sufficient background and include all relevant references? Yes:

  • Thank you very much for positive comment.

Is the research design appropriate? Can be improved:

  • Thank you very much for positive comment.

Are the methods adequately described? Can be improved:

  • Thank you very much for positive comment.

Are the results clearly presented? Can be improved:

  • Thank you very much for positive comment.

Are the conclusions supported by the results? Can be improved:

  • Thank you very much for positive comment.

Are all figures and tables clear and well-presented? Can be improved:

  • As requested a representative gel was added for generated data.

The present article 'Population genetic structure of historic olives (Olea europaea subsp. europaea) from Jordan' deals with a genetic structure analysis of 63 olive tree landraces from Jordan, with some reference cultivars. They seem to achieve some results regarding genetic structure of the olive material (K=2 and K=3 seems the better to me).

  • Thank you for the positive comment.

I think you could link your results with those presented in wider landrace sampling in the Mediterranean Basin (see Olive domestication and diversification... by Diez et al., 2015).

  • Thank you very much for turning our attention to the mentioned article, which is crucial to mention in our study to show the research gap for the historic olives from Jordan. Requested citation was made with deep thanks.

There are even more modern literature on the subject. It seems in general that all Med. landraces may be classified in three genetic groups (east, the first; west, and central).

  • Additional modern articles were added and the major missing of historic genotypes were highlighted.

Since you are presenting results of the accessions of a relatively small country, I think the relation genotype-phenotype would be interesting to present in this study. For example, fruit size, oil content, maturity date, flowering date, etc. I would consider a better accomplised study if you did so.

  • You are absolutely right. Concerning the phenotypic data, we mentioned in the revised manuscript that these are historic too old trees without any ancestry data. These trees comprise around two thirds of grown olives in our region and they represent ancient uncovered potential for the future. Therefore, the first step was to scan for their genetic structure; are they the same or they differ genetically, therefore a large sample was investigated. The attained results of the current study will be applied to the next phase study, where phenotypic data will be assessed including oil quality and characteristics.

Minor mistakes: l. 169, insert 'a' between addition and third. l. 274 , ... the Mediterranean are expected...'. l. 439 'Olea europea' in italics.

  • Thank you for your comments, which were corrected in the revised manuscript.

I would consider this document to be published in this journal if the authors introduce the phenotypic data of the landraces, and try to relate genotype and phenotype.

  • Again concerning the phenotypic data, we mentioned in the revised manuscript that these are historic too old trees without any ancestry data. These trees comprise around two thirds of grown olives in our region and they represent ancient uncovered potential for the future. Therefore, the first step was to scan for their genetic structure; are they the same or they differ genetically, therefore a large sample was investigated. The attained results of the current study will be applied to the next phase study, where phenotypic data will be assessed including oil quality and characteristics.

Comments on the Quality of English Language

In general, the English of this study is good.

  • Thank you for your positive comment.

Round 2

Reviewer 1 Report

Comments and Suggestions for Authors

The manuscript is still very weak and not ready for publication in its current state. The authors did not focus on the required revisions.

Author Response

We highly appreciate the reviewer concern regarding the absence of phenotypic data in the current investigation at this time. However, we would like to clarify that our primary objective at this stage was to conduct a broad genotypic screening of historic olive trees using ISSR markers, in order to explore their genetic diversity and structure across diverse agro-ecological zones. We are dealing with individual trees grown either in mountains or in valleys, in different orchards facing either north, south, east or west sides.

The trees investigated are centuries old (and they hugely vary, we have 200, 300, 400, 500, 600 and more years old individual trees; additional images were added to the manuscript). They are growing under markedly different environmental and management conditions, which can confound phenotypic interpretation at this stage. This is not to forget the huge effect of alternate bearing in olives especially in old trees; some will be on, others will be off.  Including phenotypic characterization without clear genetic delineation might have introduced environmentally induced noise and not to forget the climate change during the last ten years in the region. Collectively these factors would potentially have a masking effect on any meaningful genetic relationships.

By identifying individual olive tree accessions with unique and representative genotypes through genetic clustering (K = 2–8), we now have a strong foundation to select informative subsets for future, targeted phenotypic evaluation under more controlled or comparative conditions, which implies vegetative propagation of new trees for this selection and growing them in ONE stand under similar environmental conditions and management practices. This method would ensure a more reliable trait-genotype association study in the future, obviously without compromising the genetic integrity of the dataset, which was also noted in the revised manuscript.

This stepwise approach — first genotype, then phenotype — is widely recognized in germplasm conservation and utilization frameworks, especially in long-lived perennial crops like olive. Please see the 263-times cited work of Diez et al (2015), which is recognized as the corner stone and prime reference for genetic diversity of olive tree germplasm in the Mediterranean basin.

Reviewer 2 Report

Comments and Suggestions for Authors

I have no more questions. 

Author Response

We highly appreciated your positive reply.

Reviewer 3 Report

Comments and Suggestions for Authors

Dear sir, 

the paper 'Population genetic structure of historic olives (Olea europaea subsp. europaea) from Jordan' is well written but major flaws are encountered. The main one is that they don´t have phenotypic data from the studied accessions (height of the tree, flowering date, oil quality of the fruit, size of the fruit, ratio flesh-seed, etc.). I understand that in a bigger genetic analysis of the Mediterranean basin or worldwide study, the genetic structure and its discussion is enough. But not in this national study, with few accessions that cluster in only few groups. Something is missing in a manuscript intended to be published in this journal.

Besides, authors must connect the genetic group of the Jordanian varieties with the Mediterranean ones. I think they have done it, but they have to deepen more on this aspect. Jordan is near the places where olive tree was domesticated for the first time.

Therefore, I recommend rejection of this manuscript. This is my opinion.

Best

Author Response

(The authors gave the same response as above.)
